# Evolutionary history of Coleoptera revealed by extensive sampling of genes and species

Shao-Qian Zhang[1], Li-Heng Che[1], Yun Li[1], Dan Liang[1], Hong Pang[1], Adam Ślipiński[2] & Peng Zhang[1]

Beetles (Coleoptera) are the most diverse and species-rich group of insects, and a robust, time-calibrated phylogeny is fundamental to understanding macroevolutionary processes that underlie their diversity. Here we infer the phylogeny and divergence times of all major lineages of Coleoptera by analyzing 95 protein-coding genes in 373 beetle species, including ~67% of the currently recognized families. The subordinal relationships are strongly supported as Polyphaga (Adephaga (Archostemata, Myxophaga)). The series and superfamilies of Polyphaga are mostly monophyletic. The species-poor Nosodendridae is robustly recovered in a novel position sister to Staphyliniformia, Bostrichiformia, and Cucujiformia. Our divergence time analyses suggest that the crown group of extant beetles occurred ~297 million years ago (Mya) and that ~64% of families originated in the Cretaceous. Most of the herbivorous families experienced a significant increase in diversification rate during the Cretaceous, thus suggesting that the rise of angiosperms in the Cretaceous may have been an 'evolutionary impetus' driving the hyperdiversity of herbivorous beetles.

---

[1] State Key Laboratory of Biocontrol, College of Ecology and Evolution, School of Life Sciences, Sun Yat-Sen University, Guangzhou, 510006, China. [2] Australian National Insect Collection, CSIRO, GPO Box 1700, Canberra, ACT 2601, Australia. Correspondence and requests for materials should be addressed to A.Ś. (email: Adam.Slipinski@csiro.au) or to P.Z. (email: alarzhang@gmail.com)

Coleoptera, also known as beetles, are the most diverse and species-rich insect group on Earth. With more than 380,000 described extant species[1], beetles constitute ~25% of all described animal species on this planet, and many species remain to be described[2]. Beetles exhibit extraordinary morphological and ecological diversity and play important roles in nearly all terrestrial and freshwater ecosystems[3]. To understand the processes that have resulted in the extraordinary diversity of beetles, a comprehensive time-calibrated phylogeny of extant beetles is required. However, resolving the phylogeny of beetles has proven to be a difficult challenge because of their exceptional species richness, complicated morphological characteristics and sparse molecular data. Therefore, deciphering the evolutionary history of beetles is one of the most important and complicated problems in insect biology.

Since the first natural classification of beetles proposed by Crowson[4], many efforts have been made to refine the framework based on morphological characters[1,5–11]. In recent years, many researchers have attempted to resolve the beetle phylogeny on the basis of molecular data[12–17]. These studies have made great progress; however, the resolution of the resulting phylogenies is often poor, and many branches of the beetle tree-of-life remain unresolved. For example, nine hypotheses regarding the relationships among four suborders of beetles (Fig. 1) have been proposed in recent decades, but most of them did not receive strong support[11,16,18–23]. In addition, the relationships among the series and superfamilies of Polyphaga have lacked consistently strong nodal support, including the phylogenetic position of the elusive Nosodendridae. These uncertainties have prevented the development of a comprehensive time-calibrated phylogeny of beetles, which is necessary to understand the macroevolutionary processes that promoted the beetle's extraordinary diversity.

Although early beetle fossils are rare, beetles are commonly thought to have first appeared in the Early Permian[5,24]. A recent study has reported a fossil beetle from the Pennsylvanian (Carboniferous)[25], thus suggesting an earlier origin of beetles, although other researchers have suggested that the assessment of this fossil should be re-evaluated[26]. On the other hand, molecular studies have suggested that the age for crown Coleoptera ranged from ~253 to 333 Mya (million years ago) and that the divergences of most modern lineages occurred during the Late Triassic to Cretaceous; however, the confidence intervals of these age estimates are large[12,13,16,27].

It is well known that both taxon sampling and gene sampling can affect the accuracy of phylogenetic reconstruction. In previous beetle phylogenetic studies, when the amount of sequence was large (e.g., using ribosomal protein genes extracted from EST data[28], whole-mitochondrial genomes[15,17], or transcriptome data[23,29]), the taxon sampling was small, or when the taxon sampling was large, the amount of sequence was small (e.g., three genes: ~3000 nt[12], four genes: 6600 nt[14], and eight genes: 8377 nt[16]). Until recently, beetle molecular phylogenetics has mainly relied on nuclear ribosomal DNA and mitochondrial gene sequences. These data are either too conservative (lacking information) or too heterogeneous in composition and evolutionary rate (prone to systematic bias), and hence are insufficient for resolving the higher level phylogeny of beetles. Compared with nuclear ribosomal DNA and mitochondrial genes, nuclear protein-coding (NPC) genes are more informative and less biased in base composition, and they have been used to resolve many problematic relationships in beetles[16,30]. However, because of the deep evolutionary divergences in beetles and widely varying evolutionary rates among taxa, NPC genes that can be amplified across all beetles are still scarce (fewer than 10), thus resulting in their relatively infrequent use in higher level beetle phylogenetics[16].

In this study, we dramatically increase the gene sampling by including 95 recently developed[31] nuclear protein-coding genes, representing the largest source of data for beetle phylogenetics to date. In addition, our broad taxon sampling includes 373 beetles representing all recognized suborders, series, superfamilies and 124 of 186 families. Our results establish the phylogenetic relationships among major lineages and greatly improve robustness throughout the entire phylogeny, especially in deep nodes. This study provides a basis for a more accurate natural classification of Coleoptera. Furthermore, we also estimate the divergence times for the entire beetle phylogeny and study the tempo and pattern of diversification of beetles. We find that Coleoptera originated in

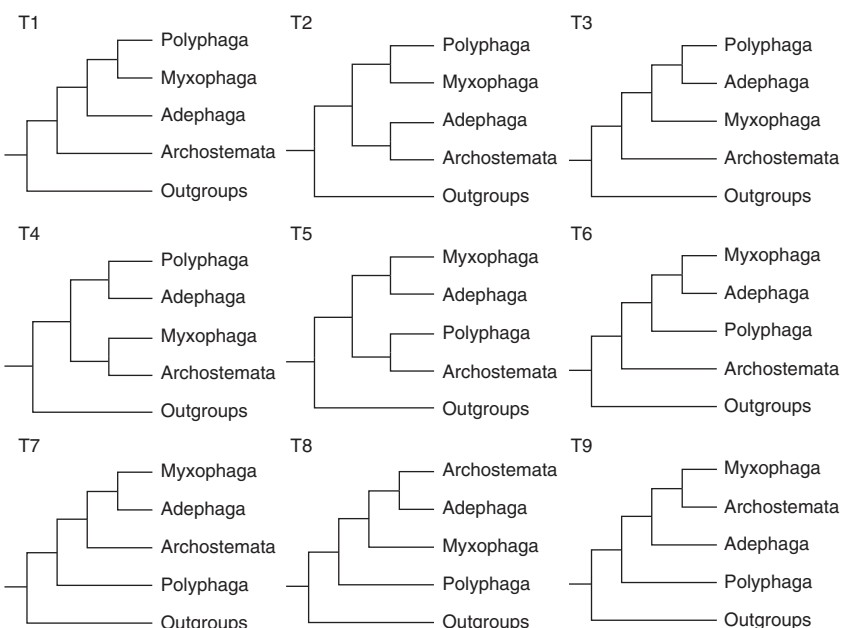

**Fig. 1** Nine proposed topologies among four suborders of Coleoptera. Topologies are derived from: T1, refs. [5,19]; T2, ref. [11]; T3, refs. [14,20]; T4, ref. [12]; T5, ref. [21]; T6, ref. [22]; T7, ref. [18]; T8, refs. [17,23]; T9 refs. [16,32].

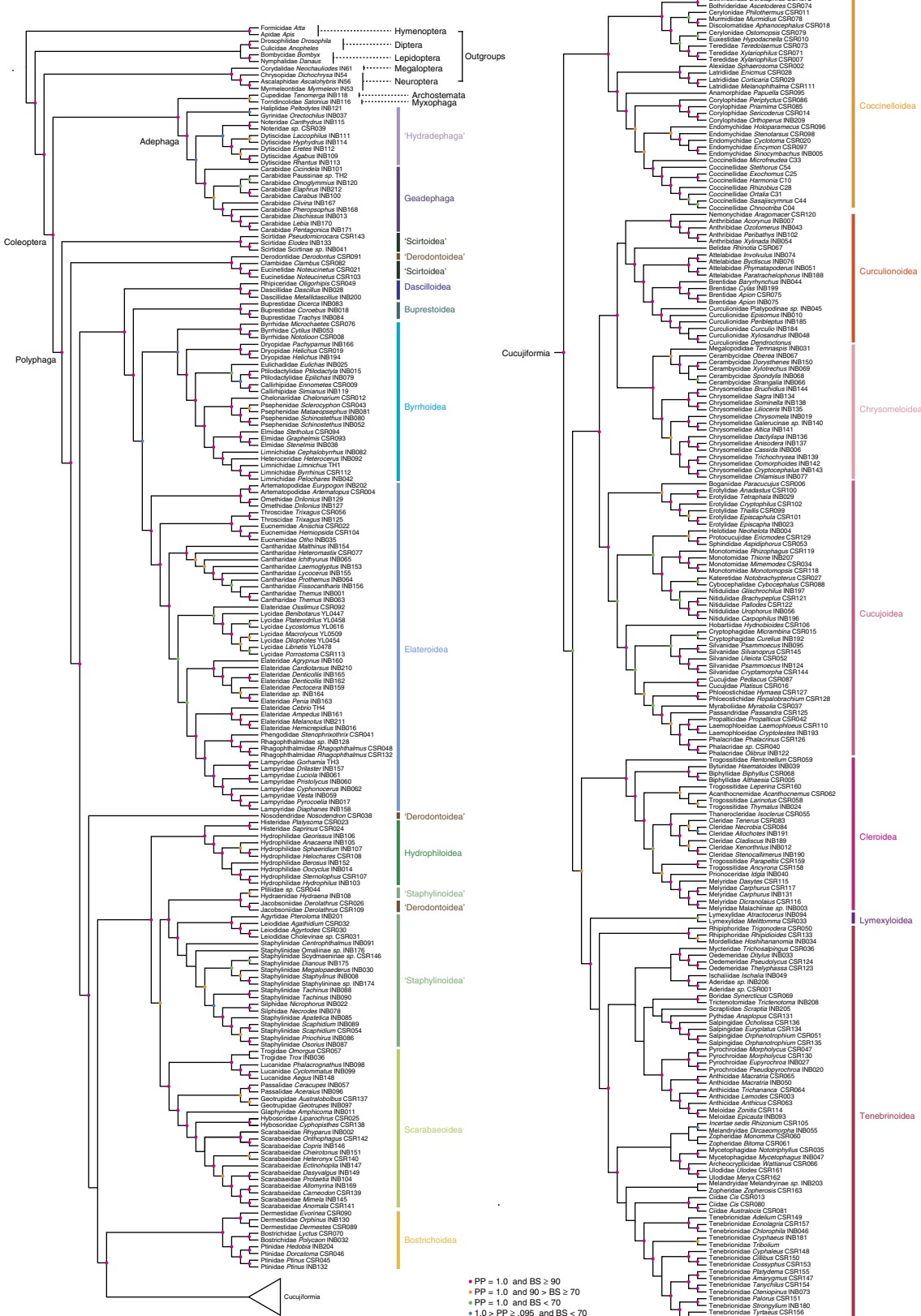

**Fig. 2** Cladogram of Coleoptera. The topology was inferred from the concatenated amino acid data set by ML (RAxML) and Bayesian inference (Exabayes). Nodes with bootstrap values (BS) <70 and posterior probabilities (PP) <0.95 are not indicated by colored circles. Traditional taxonomic units that are not monophyletic are indicated with quotation marks

**Table 1 Topology test for the nine proposed hypotheses among the four suborders of beetles**

| No. | Topology | Amino acid data | | Nucleotide data | |
|---|---|---|---|---|---|
| | | ΔIn L | AU test *p*-values | ΔIn L | AU test *p*-values |
| T9 | (Polyphaga, (Adephaga, (Archostemata, Myxophaga))) | 0 | 0.920 | 0 | 0.946 |
| T8 | (Polyphaga, (Myxophaga, (Adephaga, Archostemata))) | 23.65 | 0.123 | 16.64 | 0.151 |
| T7 | (Polyphaga, (Archostemata, (Adephaga, Myxophaga))) | 19.95 | 0.227 | 21.04 | 0.088 |
| T1 | (Archostemata, (Adephaga, (Myxophaga, Polyphaga))) | 117.83 | 2e − 05* | 90.38 | 0.002* |
| T3 | (Archostemata, (Myxophaga, (Adephaga, Polyphaga))) | 109.62 | 0.001* | 70.81 | 0.018* |
| T6 | (Archostemata, (Polyphaga, (Myxophaga, Adephaga))) | 59.96 | 0.036* | 54.42 | 0.030* |
| T2 | ((Archostemata, Adephaga), (Polyphaga, Myxophaga)) | 118.50 | 4e − 05* | 93.51 | 1e − 22* |
| T4 | ((Archostemata, Myxophaga), (Adephaga, Polyphaga)) | 92.38 | 5e − 04* | 64.45 | 0.007* |
| T5 | ((Archostemata, Polyphaga), (Adephaga, Myxophaga)) | 92.30 | 2e − 05* | 80.11 | 0.003* |

* *p*-value <0.05 indicates statistical rejection

the earliest Permian and that most extant lineages, especially phytophagous beetles, diverged during the Cretaceous, thus suggesting that the rise of angiosperms in the Cretaceous may have played an important role in the hyperdiversification of beetles. Our study provides a temporal perspective for understanding the evolutionary history of the Coleoptera and should provide a cornerstone for the further study of systematics of this extraordinarily diverse order.

## Results

**Higher level phylogenetic relationships of beetles.** Our molecular data included 95 nuclear protein-coding genes from 373 beetle species and 10 holometabolan outgroups (Supplementary Data 1). The gene coverage for species ranged from 49.1 to 96.3%, with an average of 78.6% (Supplementary Data 2). The concatenated supermatrix consisted of 23,802 amino acids (or 71,406 nucleotides). We estimated concatenated trees with both nucleotide and deduced protein sequences by using two maximum likelihood (ML) methods (RAxML and IQ-TREE) and a Bayesian approach (ExaBayes). The protein RAxML analysis produced a well-resolved phylogeny (Supplementary Fig. 1). Other ML and Bayesian analyses based on nucleotide or protein sequences resulted in nearly identical phylogenies and similar branch support, as shown in Fig. 2 (Supplementary Figs. 2–6). Gene tree-based coalescent analysis (ASTRAL) of our data recovered a tree with lower nodal support that was also congruent with the ML tree after collapsing of clades with <50% bootstrap support (Supplementary Fig. 7). These congruent results indicated that the resulting phylogeny was highly robust regardless of the data set and tree-building method.

Our phylogenetic analyses achieved well-supported resolution of relationships among all major lineages of beetles. In the ML analyses, >85% of nodes were supported with standard bootstrap values (RAxML) ≥70% or ultrafast bootstrap values (IQ-TREE) ≥95% (Supplementary Figs. 1–4). More than 95% of nodes have posterior probabilities between 0.95 and 1 in the Bayesian analyses (Supplementary Figs. 5 and 6). For this discussion, we use the protein RAxML tree as our preferred result (Fig. 2).

At the base of the Coleoptera tree, our phylogeny strongly supported the relationships among suborders as Polyphaga (Adephaga (Myxophaga, Archostemata)), a hypothesis recently reported by McKenna et al.[16] and Sharkey et al.[32], albeit with negligible to moderate support. The topology recovered in this study with high support provided an opportunity to assess relationships between suborders, which have been disputed for decades. The approximately unbiased (AU) test analysis showed that the three hypotheses with Polyphaga sister to the other three suborders were significantly better than the other hypotheses, although these three hypotheses were not significantly different

from one another (Table 1). It should be noted that our study included only a single species of Archostemata and only one of Myxophaga, which makes these taxa prone to long-branch attraction (LBA). Therefore, the relationships among Adephaga, Myxophaga, and Archostemata reported here should still be considered as tentative and needed validation by future studies.

The Adephaga has been divided into two monophyletic groups (the terrestrial Geadephaga and aquatic Hydradephaga) by many molecular studies[12,16,33]; however, Hydradephaga is sometimes recovered as paraphyletic[14,17,34]. In agreement with the latter, our analyses recovered Hydradephaga as paraphyletic with the aquatic Haliplidae and Gyrinidae forming a clade sister to all other aquatic and terrestrial adephagans (Fig. 2). However, this result had strong support only in the Bayesian analyses (BPP = 0.96; Supplementary Figs. 5 and 6) and received negligible support in the ML analyses (BS <50%; Supplementary Figs. 1 and 2). Therefore, the phylogeny of Adephaga still remained ambiguous. Given that our sampling of Hydradephaga was insufficient, recovering internal relationships of Adephaga may require sampling more aquatic adephagan families Amphizoidae, Aspidytidae, Hygrobiidae, and Meruidae.

At the base of the strongly supported suborder Polyphaga, four families occupied the basal nodes, forming two successive branching clades sister to the remaining polyphagans (Fig. 2), which is congruent with all recent studies[12,14,16,28]. In addition, we corroborated the placement of Derodontidae as the sister taxon to Clambidae + Eucinetidae with strong support (BS = 98%, BPP = 1.0; Fig. 2), which had been only weakly supported before[16].

Series Elateriformia, which consists of the monophyletic superfamilies Dascilloidea, Buprestoidea, Elateroidea, and Byrrhoidea, was strongly recovered as the third branching lineage of Polyphaga, and a similar result has only recently been proposed from an analysis of beetle mitogenomes[17].

Dascilloidea was strongly corroborated as the sister taxon to the other superfamilies of Elateriformia (BS = 100%; Fig. 2), in agreement with recent studies[14,16,35] but not with others[12,17]. Buprestoidea was recovered as a sister to a clade of Byrrhoidea and Elateroidea. However, the sisterhood between Byrrhoidea and Elateroidea was only strongly supported in Bayesian analyses (Supplementary Figs. 3 and 4), similarly to the monophyly of Byrrhoidea, in which moss-feeding Byrrhidae was only weakly related to other byrrhoids (BS = 40%; Supplementary Fig. 1).

Nosodendridae has previously been placed in various positions: within Bostrichoidea[7], within Derodontoidea[11], associated with Scirtoidea[11,36], sister to Scarabaeiformia[14], or grouped with Elateriformia[12,16]. This family was robustly recovered in a novel position in this study: as a sister clade to Staphyliniformia, Bostrichiformia, and Cucujiformia (BS = 100%; BPP = 1.0; Fig. 2).

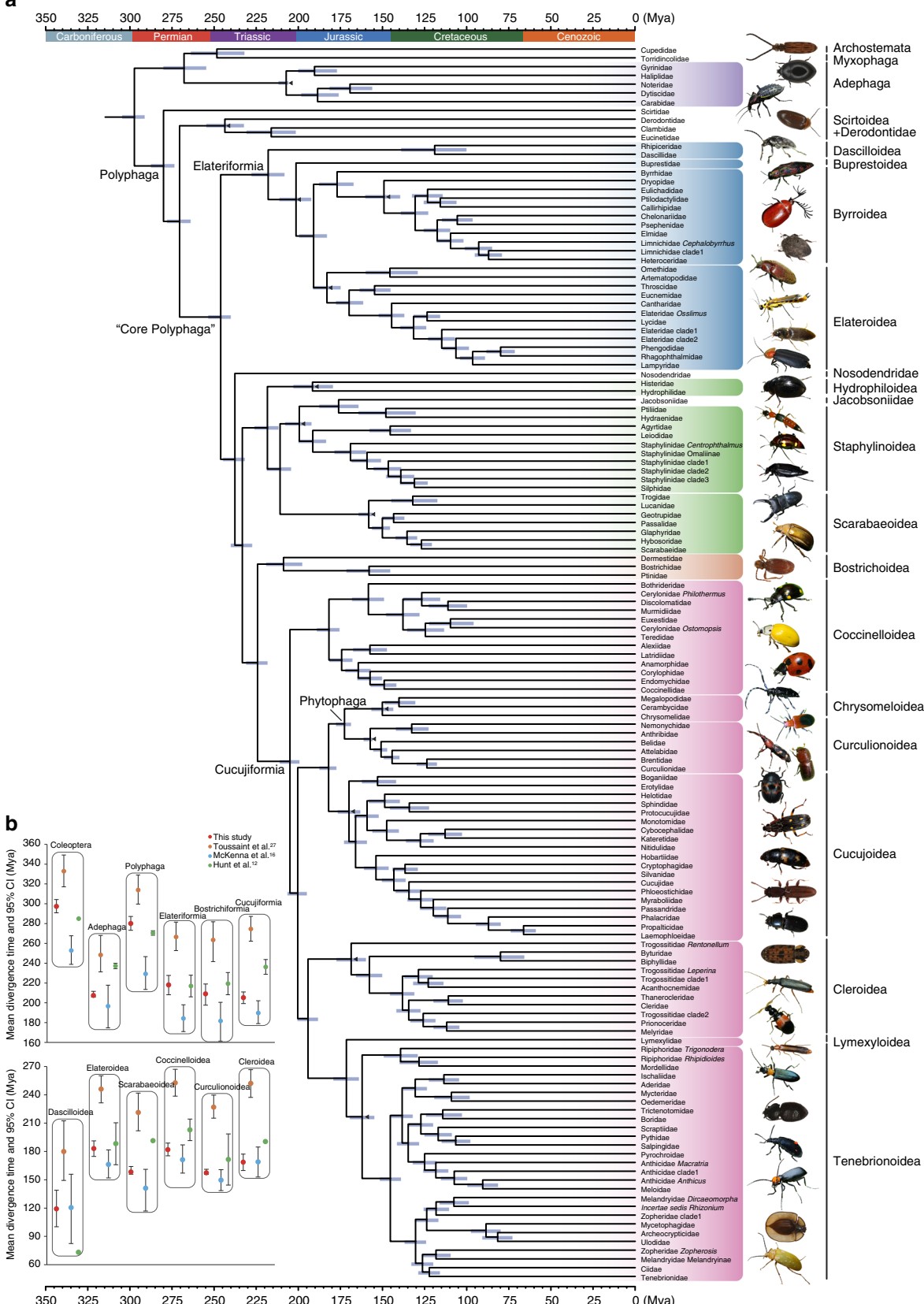

**Fig. 3** New timescale for beetle evolution and comparison of divergence times. **a** Time-calibrated tree of beetles. The time tree was collapsed to family level with outgroups removed (for detailed results, see Supplementary Fig. 8). Divergence times were estimated with MCMCTREE with 20 calibration points, on the basis of the amino acid data set. Fossil constraints within Coleoptera are shown with black triangles. Horizontal bars represent 95% credibility intervals. **b** Comparison of divergence time estimates for twelve major nodes sharing across four beetle time trees. The circle represents the mean age, and the whiskers mark the 95% credibility internals (Photo credits: Hong Pang, Yun Li, and Zhenhua Liu)

In agreement with many previous studies[16–18,37,38], Hydrophiloidea, Staphylinoidea (including Jacobsoniidae), and Scarabaeoidea formed a well-supported clade (BS = 94%; Fig. 2), thus supporting a traditional monophyletic Haplogastra. Within this clade, Staphylinoidea (including Jacobsoniidae) was closer to Scarabaeoidea than to Hydrophiloidea with moderate support (BS = 78% and BPP = 1.0; Supplementary Figs. 1 and 5), inconsistent with previous findings that Staphyliniformia (Staphylinoidea + Hydrophiloidea) was monophyletic or Hydrophiloidea was the sister group of Scarabaeoidea[8,16,37,39]. Derolathrus (Jacobsoniidae), which has previously been assigned to the superfamily Derodontoidea, was strongly recovered as sister to Hydraenidae + Ptiliidae within Staphylinoidea (BS >90%; Fig. 2). The same or similar placements were recovered on the basis of both morphological and molecular data[11,16], thus strongly suggesting that Derolathrus (Jacobsoniidae) should be transferred to Staphylinoidea.

Bostrichiformia was strongly supported as the sister group of the hyperdiverse series Cucujiformia in all of our analyses (BS = 100%; Fig. 2). Among the seven recognized superfamilies within Cucujiformia, Coccinelloidea was sister to the remaining superfamilies with moderate nodal support (BS = 69%; Fig. 2). The superfamily Cucujoidea (excluding Biphyllidae and Byturidae) was a strongly supported clade (BS = 93%; Fig. 2) and a sister taxon to Phytophaga, which consists of the two highly supported superfamilies Curculionoidea and Chrysomeloidea (Fig. 2). The close relationships among Cucujoidea, Curculionoidea, and Chrysomeloidea had maximal support in all analyses (BS = 100%; BPP = 1.0; Fig. 2). Cleroidea was strongly recovered as monophyletic (BS = 100%; Fig. 2), when including Biphyllidae and Byturidae, which were formerly placed in Cucujoidea but recently transferred to Cleroidea[12,40]. The monophyly of Lymexyloidea was moderately supported (BS = 64%; Fig. 2), and it was closely related to Tenebrionoidea with maximal support. These two superfamilies jointly are sister to Cleroidea, with moderate support (BS = 72%; Fig. 2).

In summary, our beetle phylogeny corroborates many of the deeper coleopteran nodes inferred by other studies but with greater support. Relationships among the deepest branches in the Polyphaga, for which previous studies have reported conflicting results, are now strongly supported. Our novel findings include the isolated position of Nosodendridae and a close relationship between Scarabaeoidea and Staphylinoidea.

**New timescale for beetle evolution.** Before this study, only three comprehensive family-level studies have been performed to estimate divergence times for Coleoptera with newly generated molecular data[12,13,16]. These three studies have suggested that the last common ancestor of Coleoptera first occurred in the Permian period (253–285 Mya). However, certain time estimates have been criticized by Toussaint et al.[27] because they conflict with current knowledge of the beetle fossil record. Using the data of McKenna et al.[16] but a different set of fossil calibration points, Toussaint et al.[27] have proposed a much older timescale for Coleoptera for both deeper and shallower nodes. Their results have indicated that the crown age of Coleoptera was ~333 Mya, which is in the mid Carboniferous.

The extensive sampling of nuclear genes in our study provides substantial new molecular data to estimate the divergence times for extant beetles. Our divergence time analyses used a Bayesian relaxed clock method (MCMCTREE) and 20 fossil calibration points carefully selected from currently known Coleoptera fossils (Supplementary Table 1). It should be pointed out that we used some fossils to calibrate the crown groups of the superfamilies in which they belong, even when the cited reference clearly places the fossil in extant families. We have several considerations for

doing so: (1) poor fossil preservation of beetles often prevents observation of the relevant characters, so it is possible to erroneously place fossil taxa in extant families based on incomplete morphological characters; (2) our taxon sampling does not cover all beetle families and some families are represented by only one species, the monophyly of some families is not certain yet; (3) the monophyly of most superfamilies are robust but the family-level relationships within each superfamily is not robust. Therefore, it is more proper to use these fossils at superfamily level but not family-level under the current situation of both taxon sampling and phylogenetic robustness. We also ran a time analysis using those fossils at family level to calibrate the stem ages of relevant families. The resulting times were on average 12% older than the times estimated by imposing fossils at superfamily level. This result indicated that the divergence times of beetle evolution are sensitive to the fossil calibration points, as recently suggested by Toussaint et al.[27]. Because the use of fossils in beetle divergence time analyses is still under debate[16,27], we tentatively used the divergence times estimated by imposing fossils at superfamily level as our preferred time results. The full-time tree for the 383 taxa sampled in this study is given in Supplementary Fig. 8, and the family level time tree is summarized in Fig. 3a. The 383-taxa time tree using fossils at family level can be found in Supplementary Fig. 9.

Overall, our divergence times were notably more precise (i.e., smaller confidence intervals) than those in the three previous studies. The 95% confidence intervals (CIs) for the 12 selected nodes in the beetle tree (Fig. 3b) were ~50% narrower than previous estimates. For example, the crown nodes of Coleoptera, Adephaga, and Polyphaga in our study had 95% CIs ranging from 5.9 to 14.0 Mya, whereas McKenna et al.[16] has reported CIs ranging from 28.9 to 43.1 Mya (Supplementary Table 2). We redid the time analyses using our 95 genes but the exact same age constraints as used by McKenna et al.[16] or Toussaint et al.[27] and still observed the increased precision of the dating results (Supplementary Fig. 10). This result indicated that the greater precision should be mainly attributed to the size of our data set (71 kb), which exceeds those of previous studies (~8.4 kb at most) by at least eightfold. A similar increase in precision of divergence time estimations was also found for plethodontid salamanders by using 95 nuclear genes[41].

We estimated that the last common ancestor of extant beetles occurred during the earliest Permian at 297 Mya (95% CI 291–304), which is earlier than the Early Late Permian origin (253–285 Mya) estimated by Hunt et al.[12] and McKenna et al.[16] but later than that of Toussaint et al.[27] (~333 Mya) (Fig. 3b). The initial diversification of Polyphaga occurred at 280 (273–287) Mya, in the Early Permian, but the 'core Polyphaga' (excluding basal Scirtoidea and Derodontidae) occurred at 246 (240–253) Mya, which was shortly after the Permian-Triassic (PT) boundary (Fig. 3a). Notably, the basal Polyphaga lineages are species-poor (only ~1055 species), whereas the 'core Polyphaga' includes ~88% of the described extant species of beetles (~340,000 species), and the branch leading to 'core Polyphaga' was long (24 million years in duration) and spanned the PT boundary (Fig. 3a). These results indicated that Polyphaga originated in the Permian and survived through the End-Permian mass extinction.

In shallower nodes, such as the origin of many beetle superfamilies, our time estimates are considerably younger than those of Toussaint et al.[27] and more consistent with the results of the two earlier studies[12,16] (Fig. 3b) and the results of other researches focused on individual clades of beetles[42,43]. For example, the crown age of Curculionoidea (weevils) estimated in this study is 157 (156–161) Mya, which is in the late Jurassic period, in agreement with the Jurassic origin proposed by Hunt et al.[12] and McKenna et al.[16,42]. Toussaint et al.[27] estimated

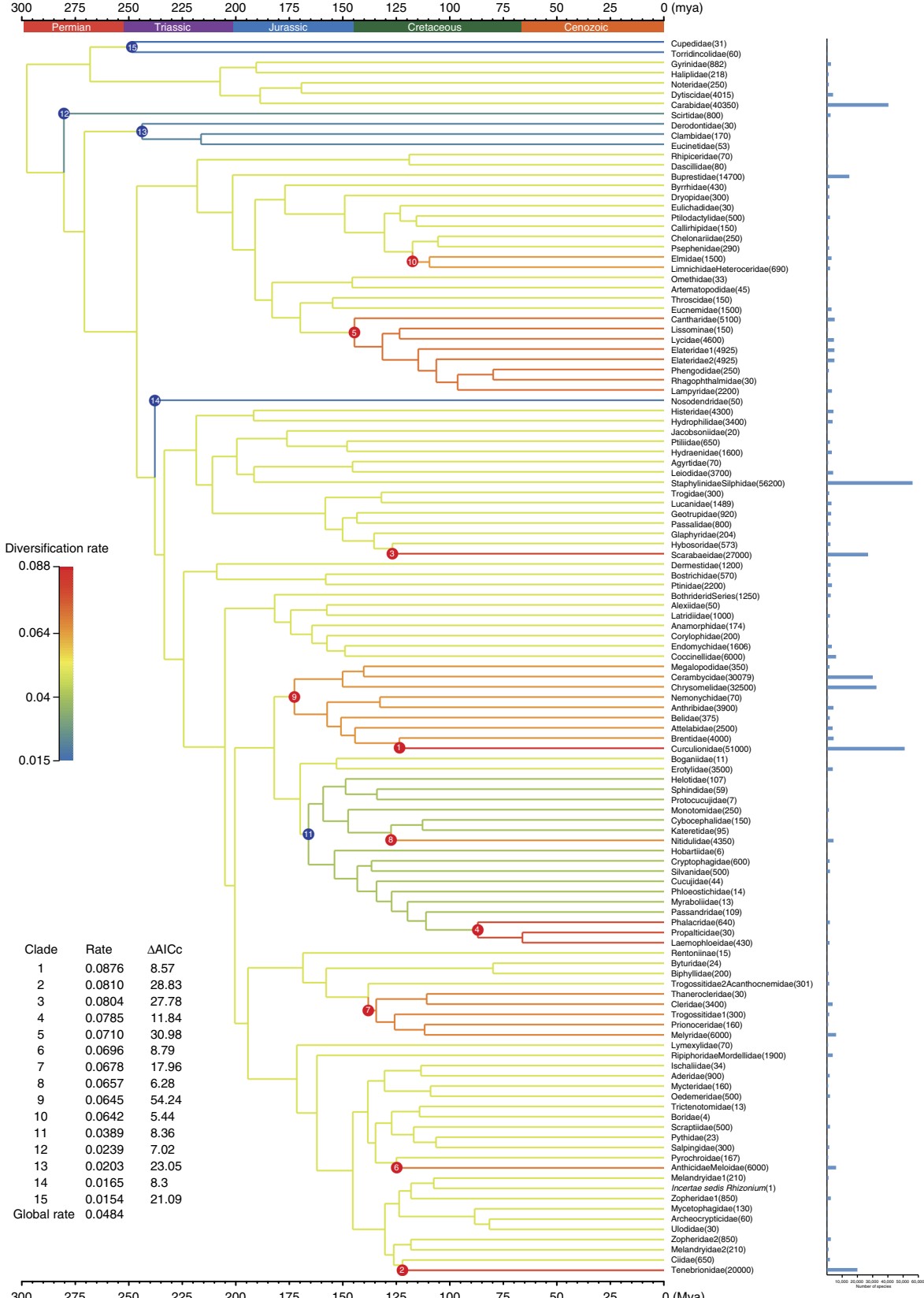

**Fig. 4** Diversification patterns of major beetle lineages inferred from MEDUSA analysis. Each terminal represents a monophyletic family-level taxon. The number in parenthesis next to the taxon name indicates the number of validly described species within the taxon. The species richness of each taxon is also indicated with histograms on the right. Branches are color coded to show the diversification rates. Clades with significant diversification rate shifts compared with the background rate are marked with circled numbers on the tree (red: rate increase; blue: rate decrease). Estimated net diversification rates and differences in AICc scores are included in the lower left table

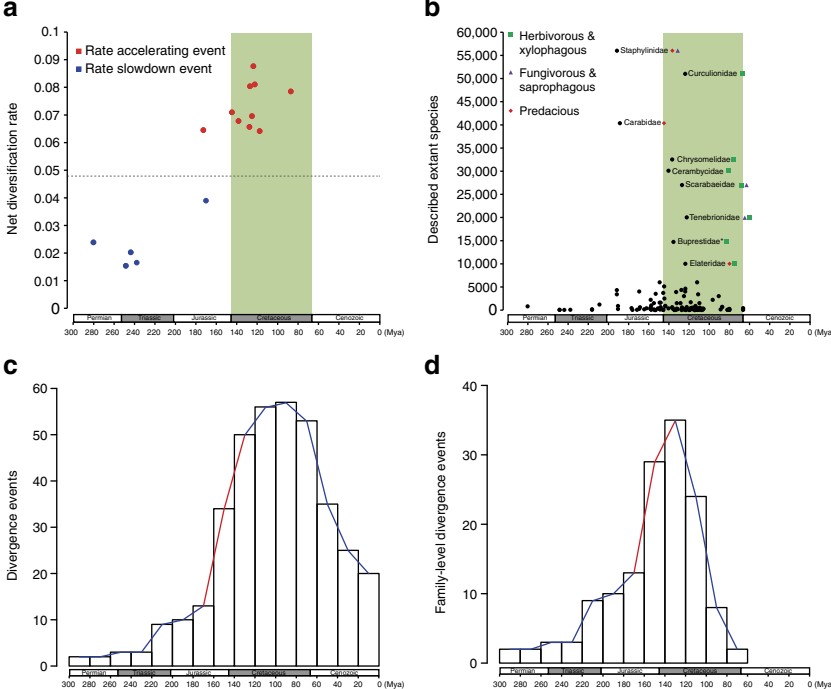

**Fig. 5** Diversification of beetles across the geological timescale. **a** Timing of the 15 significant changes in net diversification rate identified by MEDUSA. The dashed line denotes the background net diversification rate of Coleoptera. **b** Origin times and the species richness of beetle families. We used stem age as the origin time for a family when only one species is sampled for the family or when the taxon sampling did not cover the crown of the family. For non-monophyletic families, the stem age of the oldest lineage of the family was used. The nine largest families with species numbers >10,000 are highlighted with family names and feeding habits. For the Buprestidae (marked with an asterisk), we performed an additional time estimation, adding the Schizopodidae sequences from McKenna et al.[16] to calculate the stem age of this family. **c** Number of divergence events within every 20 million year interval calculated from the 383-taxon time tree. Note that the diversification rate of beetles experienced an upsurge beginning from the late Jurassic (marked with a red line). **d** Number of divergence events within every 20 million year interval calculated from the family-level time tree. A similar diversification upsurge pattern was also detected in the late Jurassic

this node to be 226.9 Mya (Late Triassic). In terms of the fossil record, weevils first appear unequivocally in the Late Jurassic (Karatau, Oxfordian—Kimmeridgian, 163.5–152.1 Mya), which is close to our time estimate.

In summary, our new timescale for beetle evolution suggests that the crown Coleoptera originated in the earliest Permian. The divergence among beetle series mainly occurred during the Triassic, with most superfamilies appearing during the Jurassic, and almost 64% of families appearing in the Cretaceous (stem ages are used here because some families have only one species sampled in this study). Even when we use the alternative calibration scheme (using fossils at family level), there are still 46% of families that originated during the Cretaceous (Supplementary Fig. 9). These age estimates corroborate many of the dating results estimated by earlier studies, but with higher precision (having smaller CIs).

**Diversification tempo of beetles and its relationship with the rise of angiosperms.** What factors cause the extraordinary species richness of beetles are still widely discussed. From a perspective of morphology, the sclerotized forewings (elytra), which protect the membranous flying hindwings, may be responsible for the apparent success of beetles[3]. Moreover, other studies have also emphasized the importance of complete metamorphosis, development cycle and division of ecological niches to larvae and adults as innovations of the extraordinary diversity of Holometabola, including Coleoptera[44,45]. Another popular hypothesis suggests that the striking diversity of beetles is largely driven by co-radiations with flowering plants[42,46]. However, Hunt et al.[12]

have argued that there is no apparent association between the diversity of beetles and the diversification of angiosperms and that the extreme diversity of beetles may be explained by their long evolutionary history, high-lineage survival, and diversification in a wide range of niches.

On the basis of the new timescale of beetles, we calculated the diversification pattern of beetles with both MEDUSA[47] and BAMM[48]. Both methods produced similar results of beetle diversification. We estimated the global diversification rate for Coleoptera to be 0.0484 lineages per million years (Myr) (MEDUSA) or 0.0510 lineages per million years (BAMM) (Fig. 4, Supplementary Fig. 11). The global diversification rate for Coleoptera drops to ~0.045 lineages/Myr based on our alternative time estimate (using fossils at family level). These rates are apparently low compared with those of other organism groups that experienced rapid radiation, such as neoavian birds (0.089 lineages/Myr)[47] and angiosperms (0.077 lineages/Myr)[49]. Because beetles originated in the lowermost Permian and have an apparently low-diversification rate, we agree with Hunt et al.[12] that the high-species richness of beetles as a whole should be attributed to their long history and low-lineage extinction.

However, the MEDUSA analysis identified ten clades within Coleoptera with significantly higher diversification rates (0.0642–0.0876 lineages/Myr) than the background diversification rate, and they all belong to the suborder Polyphaga (Fig. 4). The BAMM analysis also identified four rate-increase shifts that are included within the MEDUSA results (Supplementary Fig. 11). These clades include most species-rich groups of beetles, such as Phytophaga, Scarabaeidae, Elateridae, and Tenebrionidae, which constitute ~56.3% of extant species of beetles, thus indicating that

more than half of beetle diversity can be attributed to fast diversification rates in certain clades. In addition, both BAMM and MEDUSA analyses detected two or five clades with significantly lower diversification rates than the background rate (Fig. 4; Supplementary Fig. 11), and they mainly belong to the species-poor beetle groups. Moreover, 9 of 10 rate-accelerating events detected by MEDUSA occurred in the Cretaceous, although all five rate slowdown events occurred much earlier (Fig. 5a). A similar pattern was observed when we used the alternative timescale of beetles: seven identical rate-increasing events were detected and five of them occurred in the Cretaceous, while all rates slowdown shifts predated the Cretaceous (Supplementary Fig. 12). These results suggested that the Cretaceous was an important period in shaping the extreme diversity of beetles.

Flowering plants (angiosperms) diversified quickly during the Cretaceous period and became the dominant group of plants[50,51]. Interestingly, among the nine largest beetle families, which have >10,000 described species, seven were estimated to originate in the Cretaceous, and their diets are associated with plants (Fig. 5b). Curculionidae, Chrysomelidae, Cerambycidae, and Buprestidae are phytophagous, and more than three-quarters of species in Scarabaeidae are phytophagous, whereas certain species in Tenebrionidae and Elateridae are phytophagous, and many others feed on decomposing plant materials and woody tissues. In contrast, the other two species-rich families (Staphylinidae and Carabidae) that are predominantly predacious were estimated to originate in the Early Jurassic, long before the Cretaceous (Fig. 5b). The coordination of diversification between angiosperms and phytophagous beetles, but not with predacious beetles, clearly shows that the extraordinary diversity of phytophagous beetles can be attributed to co-evolution with angiosperms.

To show the diversification tempo of beetles through time, we counted the number of divergence events of Coleoptera within every interval of 20 million years from the Permian to the present, on the basis of our 383-species time tree. We noticed that the diversification rate of beetles experienced an upsurge in the late Jurassic (~160 Mya) and reached the greatest speed in the Cretaceous (Fig. 5c). This rate-elevating pattern remained stable when we counted divergence events on the family-level time tree (Fig. 5d). Although the oldest angiosperm fossils date from the Valanginian to the Hauterivian in the Cretaceous[52,53], the crown age of the angiosperms has been estimated to be at least 160 Mya[54,55]. Therefore, the diversification rate pattern of beetles is still consistent with the beetle-angiosperm co-evolution hypothesis[46]. However, the accelerating diversification of beetles in the late Jurassic also indicates that the rapid radiation of beetles began before flowering plants flourished.

Overall, no single explanation can explain the success of the order Coleoptera. Perhaps, the Permian origin of the crown group and a long period of evolution steadily increased the diversity of beetles. Because lineage survival was high, beetle diversification entered an 'exponential' phase in the late Jurassic. The subsequent boom of flowering plants in the Cretaceous provided new ecological opportunities for phytophagous beetles, thus further promoting the biodiversity of beetles. All these factors together create the great diversity of extant beetles.

## Methods

**Taxon sampling.** In this study, we used a Coleoptera classification that incorporated results from Ślipiński et al.[1] and Bouchard et al.[10] with the exception of Rhysodidae, which is considered a subfamily of Carabidae[11], and added the recently proposed superfamilies Coccinelloidea and eight recently elevated (or re-elevated) families[40,56]. We sampled 371 coleopteran taxa representing the 4 extant suborders, 7 series, 17 superfamilies and 124 of 186 families. Four neuropterid taxa, including 3 families of Neuroptera and 1 family of Megaloptera, were used as outgroups. Most of the missing families were species-poor lineages with limited distribution. Additionally, we added 8 taxa with public genome data downloaded

from the Ensembl database (http://www.ensembl.org), including two beetles (Tribolium castaneum and Dendroctonus ponderosae) and 6 other holometabolan insects. Therefore, our final taxon sampling included 383 species (373 beetles and 10 outgroups). We did not include Strepsiptera as an outgroup to beetles, because among the 95 genes used in this study, only 44 (missing data >50%) could find orthologous sequences in the published Strepsiptera genome[57]. All specimens derived from the Biological Museum of Sun Yet-Sen University, China and Australian National Insect Collection, CSIRO, in Canberra, Australia were marked with unique numbers. The detailed information of taxonomy, locality, collector/identifier of specimens was provided in Supplementary Data 1.

**DNA sequencing.** Specimens were preserved in 95% ethanol and stored at −20 °C. DNA was extracted from the thorax muscles, legs or the entire specimen using a TIANamp Genomic DNA kit (TIANGEN Inc., Beijing, China). Voucher specimens have been deposited in the Biological Museum of Sun Yat-Sen University. Ninety-five nuclear protein-coding genes were amplified from DNA extracts by PCR using the protocol and primers described in Che et al.[31]. The amplification products were sequenced using a next-generation sequencing (NGS) strategy, as described by Feng et al.[58]. Briefly, all amplification products from a single specimen were pooled together and purified. The specimen amplification product pools were then randomly sheared to small fragments (200–500 bp), the ends were repaired, and a species-specific barcode linker was added. All indexed amplification product pools were then mixed together, and a sequencing library was constructed with the pooled DNA using the TruSeq DNA Sample Preparation kit and sequenced on an Illumina HiSeq 2500 sequencer. Approximately 24 GB of 90-bp Illumina HiSeq paired-end reads were obtained. These reads were bioinformatically sorted by barcode sequences and assembled into consensus sequences using Trinity[59]. All assembled sequences were checked for possible intron insertion by using a Python script provided by Che et al.[31]. The final intron-removed sequences were further examined for frame shifts and stop codons to ensure that they could be properly translated. GenBank accession numbers for the new sequences are given in Supplementary Data 3.

**Sequence alignment and data partition.** All 95 genes were aligned using the ClustalW algorithm implemented in MEGA v6[60] on the basis of the translated amino acid sequences. Ambiguously aligned regions were trimmed using Gblock v.0.91b[61], with all gaps allowed (-b5 = a) and all other parameters at default settings. Nucleotide alignments were performed according to the corresponding protein alignments using a custom Python script. All 95 protein and nucleotide alignments were concatenated. Binning genes into 'supergenes' is a statistical technique that can account for sampling error by increasing signal-to-noise ratio, and it has been applied in phylogenetics recently[62–64]. Because many genes in our data set are short, we thus used data binning strategy to partition our data. The protein data set was divided into 10 partitions according to the evolutionary ratio of each gene (measured as their overall mean P distances). ProtTest 3[65] was used to identify the best-fit models for the 10 partitions with the Bayesian information criterion (BIC). We also did additional phylogenetic analyses using different partitioning schemes. The phylogenies inferred from unpartitioned and 95-gene-partitioned data set are almost identical to that inferred from 10-bin-partitioned data set, except several nodes with negligible supports (Supplementary Fig. 13). This result showed that different partitioning schemes have little influence to the final phylogenetic results. For the nucleotide data set, we used a Perl script (Degen_v1_4.pl; http://www.phylotools.com) to degenerate nucleotides to IUPAC ambiguity codes for the first and third codon positions, an extension of RY coding, which can reduce the effect of nucleotide compositional heterogeneity[66]. The degenerated nucleotide data set makes synonymous changes largely invisible but non-synonymous changes largely intact, which can be supported in RAxML and IQ-TREE analyses. The resulting nucleotide data matrix was partitioned by codons (three partitions defined), and the best-fit models for every codon partition was selected with PartitionFinder[67].

**Phylogenetic analyses.** The protein and nucleotide data sets were analyzed with both ML and Bayesian inference (BI) methods. The ML analyses were conducted using RAxML v.8.0[68] and IQ-TREE[69]. Branch support in the RAxML analyses were evaluated through a rapid bootstrap algorithm (-f a option) with 500 replicates. In the IQ-TREE ML analyses, nodal support values were estimated using the embedded ultrafast bootstrap approach (UFBoot), which is computational efficient and relatively unbiased[70].

Bayesian analyses were conducted using ExaBayes v1.4.1[71]. Two Markov Chain Monte Carlo (MCMC) runs were performed with one cold chain and three heated chains (temperature set to 0.1) for 50 million generations, and sampling was performed every 1000 generations. The average standard deviation of split frequencies (ASDSFs) and potential scale reduction factor (PSRF) were <1% and close to 1 across the two runs, respectively. The effective sample sizes (ESSs) were >200 for all parameters after the first 20% of generations were discarded.

The species tree analysis without gene concatenation was performed for the protein data set using ASTRAL 4.7.6[72] under the coalescent model. For each gene, the best ML tree and 200 bootstrapping trees were inferred by RAxML under the best fitting model selected by ProtTest. The species tree analysis was then conducted using ASTRAL taking these 95 unrooted best ML trees and

corresponding bootstrapping trees as input, under the multilocus bootstrapping option with 200 replicates (-r = 200).

The approximately unbiased (AU)[73] test was used to evaluate the alternative beetle phylogeny hypotheses, and site-wise log likelihoods of all alternative topologies were calculated with RAxML using the -f g option. Then, the site log-likelihood file was used as input to estimate the P-values for each alternative hypothesis using the AU test implemented in the program CONSEL[74].

**Divergence time estimation.** Divergence times were estimated by using MCMCTREE in the PAML package[75] with the uncorrelated rate model (clock = 2). The protein RAxML tree was used as the reference topology. Twenty fossils (Supplementary Table 1), of which 13 were within the Coleoptera, were used to calibrate the clock. Imposing maximum bounds is necessary when estimating deep divergence times[76]. Because holometabolous insects are not known from before the Pennsylvanian of the Carboniferous, we constrained the maximum age of the Coleoptera-Neuropterida split to 323.2 Mya, the Mississippian-Pennsylvanian boundary, which is a fairly conservative maximum bound for this node. To further limit the effect of imposing an erroneous maximum constraint, we specified the tail probability of this maximum bound as 2.5%; thus, the time estimation had a 2.5% probability of being greater than the bound.

The ML estimates of the branch lengths for each of the 10 protein partitions were calculated using CODEML (in PAML) under the WAG + F + Γ model. To specify the prior on the overall substitution rate, the root age (crown Holometabola) was set to 345 Mya, according to a recent phylogenomic estimate[23]. On the basis of the mean tree depth from the 10 protein partitions, the gamma-Dirichlet prior for the overall substitution rate (rgene gamma) was set to G (1, 8.36), and the gamma-Dirichlet prior for the rate-drift parameter (sigma2 gamma) was set at G (1, 4.5). The posterior time estimates were conducted by using a MCMC algorithm. After the first 100,000 iterations were discarded as burn-in, the MCMC run was sampled every 100 iterations until it achieved 10,000 samples. Two MCMC runs using different random seeds were compared to determine their convergence with similar results, and effective sample sizes of every node age and every parameter were >200, as determined in Tracer V1.4[77] software.

**Diversification rate analyses.** We used MEDUSA(Modeling Evolutionary Diversification Using Stepwise AIC)[47] and BAMM (Bayesian Analysis of Macro-evolutionary Mixture)[48] to investigate the tempo of diversification of the Coleoptera. MEDUSA was conducted on the ultrametric phylogenetic tree with the species richness data. The time-calibrated Coleoptera tree generated by MCMCTREE was pruned to a family-level chronogram so that each terminal reflected a monophyletic family (or possible equal). In some cases, families that were not monophyletic were grouped together or split apart. The approximate numbers of described species for terminals were obtained from Ślipiński et al.[1]. MEDUSA sequentially adds rate shifts to the family-level chronogram until further additions fail to have a distinct increase in model fit (i.e., the improvement in AIC score is lower than the threshold). The MEDUSA analysis was conducted in R using the packages Geiger[78] with the default settings, including the corrected AIC (AICc) criterion and mixed model.

We also detected diversification rate shifts using BAMM v. 2.5. The 383-taxa time-calibrated tree was used as input tree. To account for incomplete taxa sampling, we used a non-random incomplete taxon sampling correction and specified this sampling fractions by families. The family-specific sampling probabilities were specified according to described species diversity obtained from Ślipiński et al.[1]. The BAMM analysis was run for 60 million generations at a temperature increment parameter of 0.01 and sampled event data every 1000 generations. We discarded the first 10% generations as burn-in and examined the effective sample size (ESS >200) of the log-likelihood and the number of shift events for convergence with the CODA package[79] in R. Finally, the best shift configuration and the net diversification rates of clades were inferred with BAMMtools[80].

**Data availability.** The raw sequences of the 95 genes, nucleotide and amino acid alignments of 95 genes, phylogenetic trees, and time trees are available in figshare (10.6084/m9.figshare.5306497). All new gene sequences have been deposited in GenBank (for accession numbers, Supplementary Data 3). The raw Illumina sequencing data generated in this paper can be downloaded from the NCBI Sequence Read Archive under the BioProject Accession Number PRJNA419242.

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

## Acknowledgements

We thank all of our lab members for help in experiments and data analyses. We thank Zhenhua Liu for kindly providing valuable beetle images. This work was supported by the National Natural Science Foundation of China (grants No. 31672266 and 31372172 to P. Zhang) and the National Youth Talent Support Program (W02070133 to P. Zhang).

## Author contributions

P.Z., D.L., and A.S. designed the project. Y.L., H.P., and A.S. carried out taxon sampling and collection. S.-Q.Z. performed the DNA sequencing with the help of L.-H.C. and Y.L. S.-Q.Z. and P.Z. analyzed the data. S.-Q.Z. and P.Z. wrote the paper.

## Additional information

**Competing interests:** The authors declare no competing financial interests.

