## [Peer Review File · Nature Communications]

Reviewers' comments:

Reviewer #1 (Remarks to the Author):

Review of the manuscript entitled "Evolutionary history of Coleoptera revealed by extensive sampling of gene and species" by S-Q Zhang et al.

The manuscript is a large-scale study on the phylogeny and evolution of beetles, or Coleoptera, using molecular data. Coleoptera is the most species-rich insect order in the world based on number of described species, and constitutes around 25% of all named animal species. Previous molecular studies on beetles have used a large taxon sampling but few genes, or many genes but with few terminals. This is the first study that tries to accomplish many nuclear genes (95) AND a large taxon sampling (373 terminals, representing 124 of 186 families).

The research group has in a separate paper developed a set of 95 nuclear protein coding genes based on available genomes of a few species, for use in beetle phylogenetics, and these are here used to generate the underlying dataset which is on average 78% complete in terms of gene coverage for terminals.

Developing this set of genes, and using them on a large taxon sampling for beetles is a new significant leap for beetle phylogenetics, as well as a proof-of-concept that the 95 genes developed are highly informative and efficient and basically represents an alternative to gene capture methods or transcriptomics.

Analysis-wise the study is less ground-breaking in terms of methodology and questions asked. Basically, the study revisits previous postulated and contested hypotheses, now with a better-supported backbone phylogeny of Coleoptera. That is fine – it will be the dataset and what the results tell us about the relationships and evolution in beetles that will be most reader's interests; and that backbone nodes finally come out strongly supported in a study like this.

These are my comments, small and large:

The discussion of the smaller confidence interval for inferred node dates (lines 198-206, again at line 234) compared to earlier studies, and attributing this to the larger size of the dataset is misleading without also discussing the effect of age priors (calibrations). It is true that part of the uncertainty (confidence interval) will decrease with increasing sampling of loci, but there is also another part to the uncertainty, that connected to the calibrations, which is not affected. Without comparing the calibrations between the studies it is not clear how the authors can attribute the smaller intervals to the larger data size.

I note a heavy bias in geographical origin of the sampled taxa to China and Australia. If this might bias the inferred phylogeny in some way I do not know. It is still the best-sampled phylogeny of beetles in terms of BOTH many nuclear genes AND a large taxon sampling representing a very good representation of world families (124 out of 186). But one

unfortunate effect is that some taxa that would have been useful and important are lacking, for instance Trachypachidae in Adephaga and additional representatives of Myxophaga and Archostemata.

The authors use Medusa for diversification rate analysis. I believe this method has received substantial critique (Rabosky, 2014 PlosONE; May & Moore, 2016 Syst. Biol.) and that most people now use BAMM, which itself is currently heavily debated (Moore et al 2016, PNAS; Rabosky et al 2017 Syst. Biol.). I think it would be prudent to at least compare the results from Medusa to the ones from BAMM, if not replace them.

My largest concern methodology-wise is perhaps the calibrations in the dating analysis. Here the authors use 20 "carefully selected" (line 192) fossils, 13 within Coleoptera. The line (193-195) is critical: "To avoid misplacing fossils for calibration, we conservatively used a fossil to calibrate the stem age rather than the crown age if the fossil did not definitely belong to the crown group." This sounds like a sound statement, and what I would have done as well. But then looking at supplementary table 3 with the details of the fossils used for calibration, their minimum age, a reference to the fossil (described or discussed) and how they are used (Calibrated nodes, also indicated by triangles in fig 3) I was surprised. They were not only conservatively used as calibration to crown or stem of the group in question depending on evidence of belonging in the crown. They were always used at superfamily level, even if the cited reference clearly places the fossil in extant families or even higher (subfamilies, tribes). My question is why? Why use them only at the superfamily node when they can be assigned higher up? This is in fact not a conservative approach at all but results in that the authors for instance estimate the stem age Hydrophilidae to 143My even though they cite a Hydrophilidae fossil with a minimum age of 151My, the stem age of Lucanidae to 132My while citing a Lucanidae fossil with minimum age 156My and so on (there are more examples). If the authors claim that the higher assignments made in the cited references are erroneous or are not certain then this must at least be motivated, but I doubt that this is the reason (at least I consider several of them well motivated and justified).

This is clearly an unsatisfactory approach, results in a dated tree already rejected by available fossils cited in the same study, and underestimates node ages in the tree. It may or may not affect the later results and conclusions related to the diversification rate analyses, and rate changes for herbivorous groups coinciding in time with the rise of angiosperms.

Minor things

Manuscript Title. Plural s for genes

Line 118. Wrong ref number referred to

Line 169-171. This was showed already by ref 12

Line 219-220. The first part of this sentence is showed, but that they underwent "rapid divergence after the extinction" is unclear.

Line 241 "of" lacking

Line 266 a "t" lacking in plants
Line 373 was should be is

Suppl table 4
Table heading: Sharing should be shared
Line 126 includes better than covers
Line 125 Hydrophiloidea misspelled

Suppl table 2
Is column PI Sites (Parsimony informative sites) important for anything in the study?
Parsimony analysis is not used in the study, alignment length together with number of variable sites seems sufficient.

The study by Sharkey et al (Scient. Rep. 2017), should be mentioned, for instance when discussing the relationship between the four suborders, as they found the same as in the present study. It uses 358 gene clusters from translated transcriptome assemblies to infer a species phylogeny of some 50 taxa of beetles including all four suborders, even though it has another focus than beetle phylogenetics.

Supplementary table 1 can be improved. First it is not useful for an overview to have it sorted by voucher number – better after taxonomic units. Second the locality column is inconsistent. For example country is not always given first but sometimes just QLD (for Queensland, Australia), "Thailand material" what is that?, Romania is misspelled, name of collector is often given without this being explained in header, same collector sometimes with sometimes without initials, etc

Reviewer #2 (Remarks to the Author):

Review of "Evolutionary history of Coleoptera revealed by extensive sampling of gene and species"

This is a valuable paper on the phylogeny of one of the major lineages of life. Its value relative to other recent papers is the large number of nuclear protein-coding genes (NPC) used for a relatively large number of species. Its family level coverage (67% of living families) is not as extensive as McKenna et al. (94% of living families), and has some serious deficiencies, but its NPC sampling is much more extensive (95 NPCs versus 6 NPCs).

The paper is worth publishing, but it does have some critical problems, which should be corrected.

1. Identifications of specimens.

This is handled poorly, and yet it is a key aspect of any systematics paper.

Among the most concerning problems is the lack of identifications of many of the specimens. A full 69 of the species are not identified to genus, and frequently not to subfamily. I have never seen a systematics paper with anywhere near this level of “unknowns”. The 95 genes for each of those specimens will effectively be lost to the world – they won’t be able to be combined with any data in the future in any sensible way, except at the broadest levels. (Yes, eventually someone knowledgeable might examine the vouchers, but until that happens, the data have little value.) What does one do in the future with data from some unknown Dryopidae, or unknown Cryptophagidae? It can’t possibly be used for analyses within those families, and to the extent that some families are not monophyletic, it may not be usable at higher levels either. I realize that it is hard to identify beetles to genus in some parts of the world, but surely more can be done to identify them to a finer level than just “Cryptophagidae sp.”. If some of those belong to undescribed genera, then at least give an indication as to what they are close to, e.g., “New genus near X”. If the authors do not themselves have the coleopterological knowledge to identify specimens, they should reach out to experts in various groups to help with identifications.

This doesn’t just affect future use of the data, it affects a reader’s interpretations of the results. For example, consider the family Dryopidae. If someone were interested in its placement, knowing whether a diverse sample of dryopids are included in the analysis would be important. If the authors sampled dryopids representing the diversity of the family, then I would trust the results more than if only one small group of closely related dryopids were included. But all three dryopids included are listed simply as “gen. sp.”, so we have no idea whether they represent diverse dryopids or are extremely close.

Along a similar vein, it is not clear who identified many of the taxa. Without that knowledge, we cannot judge the quality of the identifications. E.g., Who identified the *Myrabolia*? Can we trust the identification? The authors state that the collectors identified the specimens, but it is not always clear who the collectors were. In Supp Table 1, we can infer that a single word at the end of the Locality might be the last name of a collector, but this is not always clear. And if it is clear, it can require special knowledge to know of whom they speak. For example, there are numerous specimens with the word “Monteith” at the end of the locality data. I presume this is Geoff Monteith, but how are other readers to know this? The identifier should be listed, clearly, for all specimens. And not just their last names, but their full names, so that readers have the information they need to judge the quality of the identifications.

2. Some major results are based upon very few species, and could easily be artifacts resulting from long branch attraction.

A major result of the paper, perhaps the major result, are the inferred relationships of the four suborders. However, included in their analysis is one representative of Archostemata and one of Myxophaga. This is in notable contrast to McKenna et al. 2015, which had in their analysis seven species in three families of Archostemata, and six species in all four families of Myxophaga. Having only a single species of Archostemata and only one of

Myxophaga makes these taxa prone to long-branch attraction, and thus their placement is much less certain. This is especially problematic given that the relationships of the four suborders are in part an issue of rooting: the nine proposed hypotheses of subordinal relationships listed in Table 1 represent only three unrooted hypotheses. Because of the long phylogenetic distance to the outgroups, the position of the root of Coleoptera is especially sensitive to long-branch attraction; having only one archostematan and one myxophagan makes it even more so.

Another example are the claims about the paraphyly of Hydradephaga. This paraphyly results from the position of Haliplidae and Gyrinidae. However, these divergent families are represented by only a single species each, and thus they are on long branches. These long branches are more likely to be attracted to each other, and are more likely to be artificially attracted to the base of Adephaga.

I realize it would be difficult to add more Myxophaga and Archostemata at this point. If this cannot be done, then the authors need to be explicit about the problems of having only single species, including potential long branch attraction, and should therefore be less bold in their claims.

3. Timescale

The authors state "We attribute the greater precision [of dating estimates] to the size of our dataset (71 kb), which exceeds those of previous studies (~8.4 kb at most) by at least eight fold". It's not clear to me why this would necessarily follow, as there are two differences between their analysis and McKenna et al.: the amount of DNA data and different age constraints. The authors claim, if true, would be very interesting. For this reasons it would be very valuable if the authors not just speculate, but actually test this. If they redo the analysis using their 95 genes but the exact same age constraints as used by McKenna et al., and they still had the increased precision, then their claim would be corroborated; if they had the same precision as McKenna et al, then the result could be a result of the different age constraints.

Methods:

1. The authors should provide some argument as to why they did not include representatives of the sister group of beetles, Strepsiptera, as an outgroup to beetles.
2. Why was the data set divided into 10 partitions based upon rate? Why rate? Why 10? A more coherent method is to use a program such as PartitionFinder to determine the optimal partitioning of data. This will take into account factors that are more critical to model fitting than is simple rate.

Some minor issues:

The paper Baca et al. 2017 ("Ultraconserved elements show utility in phylogenetic inference of Adephaga (Coleoptera) and suggest paraphyly of 'Hydradephaga'") should be cited, and its results incorporated into the discussion.

The authors state "We estimated that the last common ancestor of extant beetles occurred during the earliest Permian at 297 mya (95% CI 291–304), which is earlier than the Early-Late Permian origin (285–253 mya) estimated by Hunt et al. and McKenna et al. but later than that of Toussaint et al. (~ 333 mya) (Fig. 3b). The earliest Permian origin of Coleoptera suggested in this study is consistent with the beetle fossil record, because the oldest definite beetle fossil (*Coleopsis archaica*) dates back to the earliest Permian at approximately 295 million years ago". The logic of the last sentence is not clear to me. *Coleopsis* is part of the stem lineage of Coleoptera, and for this reason could occur before or after the origin of the common ancestor of extant beetles; in what way is the age of *Coleopsis* relevant for the age of the common ancestor of extant beetles?

Spelling of authors name in reference 16 is incorrect.

Reference 30 has an incorrect title

Reviewer #3 (Remarks to the Author):

The authors report what is likely to be the most complete molecular dataset so far published for Coleoptera, and reconstruct a well resolved, well supported calibrated phylogeny of the Order. The methodological details were published in an associated paper (see below), but the results seem robust and the conclusions well justified. In my opinion the paper is acceptable for publication, with only some minor modifications.

A relatively high proportion of taxa is not identified not even at the genus level. The authors provide a substantial genetic resource for many taxa, but its usefulness is greatly diminished if the identification is only at the family level. In some cases it may be complex to arrive at a generic identification, but for others this seems relatively simple, as there are many genera-poor families for which the identification should be possible.

In the Abstract, the sentence "sister to all other polyphagans, excluding Scirtoidea, Derodontidae and Elateriformia" (as well as similar sentences in the text) may be confusing, as it suggests a more basal position for Nosodendrinae. Would it be better something like "sister to Staphyliniformia, Bostrichiformia and Cucujiformia"?

The sentence " nine hypotheses regarding the relationships among four suborders of beetles (Fig. 1) have been proposed in recent decades, although none have received strong support^{11,16,18–23}." is misleading, as in some cases topologies were strongly supported, only with a poor taxonomic sampling (e.g. Pons et al.).

"The inconsistency between phylogenetic relationships and variable age estimates is probably because of the small number of molecular markers traditionally used for beetle phylogenetics." - the effect of a poor sampling must also be considered. In previous phylogenies, when the amount of sequence was enough, the sampling was very poor (e.g. Misof et al.), or when the sampling was good, the amount of sequence was minimal (e.g. Hunt et al.). However, the large differences with the dating of Toussaint et al. may require some additional explanation. Could they be because of the different use of fossils? Some methodological differences in the calibration? "Small number of markers" does not seem to be reason enough here.

The choice of the partition scheme needs more explanation. Why ten partitions according to the p distances of the genes? What effect could do the use of say 9, or 11 partitions? Seems an arbitrary choice, which either needs more justification, or to be shown that the actual number is irrelevant within a reasonable range.

"For the nucleotide data set, we used a Perl script (Degen_v1_4.pl; <http://www.phylotools.com>) that degenerates the first and third codons encoding the same amino acid to minimize heterogeneous base composition⁵⁹" - the authors should provide more details. RAxML, Beast and other software cannot deal with ambiguous positions - they should be either missing (or N) or defined. Were all 1st and 3rd codon positions just transformed to "N", i.e. effectively deleted from the analyses?

There is no information of the primers used - or even the full name of the genes - other than a reference to an associated work (Che et al.). It would be useful to have more data, at least in the supplement.

In Table 1 it would be useful to include the numbers of the topologies in Fig. 1, to readily see which is which

Should be "Drosophilidae" not "Drosophylidae" (Supp. Table 3) and "Anacaena" not "Anaceana" (Fig. 2, supplements)

The alignments are said to be deposited in Dryad: I assume the sequences are also to be deposited in GenBank?

Reviewers' comments:

Reviewer #1 (Remarks to the Author):

Review of the manuscript entitled “Evolutionary history of Coleoptera revealed by extensive sampling of gene and species” by S-Q Zhang et al.

The manuscript is a large-scale study on the phylogeny and evolution of beetles, or Coleoptera, using molecular data. Coleoptera is the most species-rich insect order in the world based on number of described species, and constitutes around 25% of all named animal species. Previous molecular studies on beetles have used a large taxon sampling but few genes, or many genes but with few terminals. This is the first study that tries to accomplish many nuclear genes (95) AND a large taxon sampling (373 terminals, representing 124 of 186 families).

The research group has in a separate paper developed a set of 95 nuclear protein coding genes based on available genomes of a few species, for use in beetle phylogenetics, and these are here used to generate the underlying dataset which is on average 78% complete in terms of gene coverage for terminals.

Developing this set of genes, and using them on a large taxon sampling for beetles is a new significant leap for beetle phylogenetics, as well as a proof-of-concept that the 95 genes developed are highly informative and efficient and basically represents an alternative to gene capture methods or transcriptomics.

Analysis-wise the study is less ground-breaking in terms of methodology and questions asked. Basically, the study revisits previous postulated and contested hypotheses, now with a better-supported backbone phylogeny of Coleoptera. That is fine – it will be the dataset and what the results tell us about the relationships and evolution in beetles that will be most reader’s interests; and that backbone nodes finally come out strongly supported in a study like this.

These are my comments, small and large:

The discussion of the smaller confidence interval for inferred node dates (lines 198-206, again at line 234) compared to earlier studies, and attributing this to the larger size of the dataset is misleading without also discussing the effect of age priors (calibrations). It is true that part of the uncertainty (confidence interval) will decrease with increasing sampling of loci, but there is also another part to the uncertainty, that connected to the calibrations, which is not affected. Without comparing the calibrations between the studies it is not clear how the authors can attribute the smaller intervals to the larger data size.

Reply: We have done an additional time analyses with our data set and exact calibration points taken from other studies (McKanna et al. 2015; Toussaint et al. 2017). The confident intervals are still smaller than those from McKanna et al. and Toussaint et al. This result indicates that smaller confidence intervals resulted in this study mainly attribute to the larger size of our data. We have mentioned this new result in the revised ms with a new supplementary figure (Supplementary Fig.

10).

I note a heavy bias in geographical origin of the sampled taxa to China and Australia. If this might bias the inferred phylogeny in some way I do not know. It is still the best-sampled phylogeny of beetles in terms of BOTH many nuclear genes AND a large taxon sampling representing a very good representation of world families (124 out of 186). But one unfortunate effect is that some taxa that would have been useful and important are lacking, for instance Trachypachidae in Adephaga and additional representatives of Myxophaga and Archostemata.

Reply: Our taxon sampling is based on the long-term collection by our coauthors: Li Yun (China) and Adam Ślipiński (Australia), so it is not surprise that there is a geographical bias of our sampled beetle taxa. But we don't think this bias will affect the phylogenetic conclusions of our study because our focus is the family-level. We admit that our taxon sampling lacks some important taxa, but our sampling is still a very good representation of world beetle families. Beetles are such a huge group, so it is SO difficult to get everything you need. We hope the reviewer can understand that we have tried our best to collect beetle samples.

The authors use Medusa for diversification rate analysis. I believe this method has received substantial critique (Rabosky, 2014 PlosONE; May & Moore, 2016 Syst. Biol.) and that most people now use BAMM, which itself is currently heavily debated (Moore et al 2016, PNAS; Rabosky et al 2017 Syst. Biol.). I think it would be prudent to at least compare the results from Medusa to the ones from BAMM, if not replace them.

Reply: Thank you for your suggestion. We have redone the diversification rate analysis with BAMM as suggested. The results from BAMM are similar to the results from MEDUSA. Both methods estimate the diversification rates of beetles (MEDUSA: 0.0484 lineages/Ma; BAMM: 0.0510 lineages/Ma) slightly lower than neoaves or angiosperms. In addition, MEDUSA identified fifteen clades with significant rate shifts (rate increase: 10 clades; rate decrease: 5 clades) compared with the background diversification rate. The BAMM analysis only identified six clades with significant rate shifts, but this result is totally compatible with the MEDUSA result. We have mentioned the new diversification analysis and have added the BAMM results in the revised ms with a new supplementary figure (supplementary Fig. 11).

My largest concern methodology-wise is perhaps the calibrations in the dating analysis. Here the authors use 20 “carefully selected” (line 192) fossils, 13 within Coleoptera.

The line (193-195) is critical: “To avoid misplacing fossils for calibration, we conservatively used a fossil to calibrate the stem age rather than the crown age if the fossil did not definitely belong to the crown group.” This sounds like a sound statement, and what I would have done as well. But then looking at supplementary table 3 with the details of the fossils used for calibration, their minimum age, a reference to the fossil (described or discussed) and how they are used (Calibrated nodes, also indicated by triangles in fig 3) I was surprised. They were not only conservatively used as calibration to crown or stem of the group in question depending on evidence of belonging in the crown. They were always used at superfamily level, even if the cited

reference clearly places the fossil in extant families or even higher (subfamilies, tribes). My question is why? Why use them only at the superfamily node when they can be assigned higher up? This is in fact not a conservative

approach at all but results in that the authors for instance estimate the stem age Hydrophilidae to 143My even though they cite a Hydrophilidae fossil with a minimum age of 151My, the stem age of Lucanidae to 132My while citing a Lucanidae fossil with minimum age 156My and so on (there are more examples). If the authors claim that the higher assignments made in the cited references are erroneous or are not certain then this must at least be motivated, but I doubt that this is the reason (at least I consider several of them well motivated and justified).

This is clearly an unsatisfactory approach, results in a dated tree already rejected by available fossils cited in the same study, and underestimates node ages in the tree. It may or may not affect the later results and conclusions related to the diversification rate analyses, and rate changes for herbivorous groups coinciding in time with the rise of angiosperms.

Reply: There are some considerations for using fossils at the superfamily nodes. First, we know that morphological convergence is very common in beetles so that it is possible to erroneously assign fossils in extant families or even higher. Second, the number of described extant species of beetles is about 400,000 and our taxon sampling represents a little part of beetles; there is only one representative for some families, such as Gyrinidae or Mordellidae, and the monophyly of these families is not certain yet in this study. Third, we did not cover all extant beetle families and interrelationships between families within superfamilies are not robust, and there are much conflicts about family-level relationships between different studies (Hunt et al. 2007; Bocak et al. 2014; McKenna et al. 2015; this study). On the other hand, the monophyly of most superfamilies are robust as showed in several recent studies (McKenna et al. 2015; Timmermans et al. 2016; this study). Therefore, we think that it is more proper to use these fossils at superfamily level but not family level under the current condition of both taxon sampling and phylogenetic robustness.

We admit that some of our dating results are in conflict with known fossil records. For the Hydrophilidae, as we did not cover all genera of Hydrophilidae in this study, the crown of Hydrophilidae may be underestimated. Note that the most recent common ancestor of Hydrophilidae and Histeridae (the stem of Hydrophilidae) is 192 Ma, still older than the Hydrophilidae fossil of 151 Ma. The Lucanidae fossil *Juraesalus atavus* was described as “amongst the oldest records of Scarabaeoidea” in the cited reference (Nikolajev et al. 2011), so we conservatively assigned this fossils to the crown Scarabaeoidea and the estimated age of Scarabaeoidea is 158 Ma, older than the fossil of 156 Ma.

Echoing your concerns, we have done an additional dating analysis using those fossil ages to calibrate the stem nodes of their assigned families. The resulting node ages are on average 12% older than the ones estimated by using fossils at superfamily node. This result indicates that the divergence times of beetle evolution are sensitive to the fossil calibration points, as also observed in two recent studies that used the same data set but different sets of calibration points (McKenna et al. 2015; Toussaint et al. 2017). Because there are different opinions in the use of fossils for calibration (McKenna et al. 2015; Toussaint et al. 2017), and the divergence times of beetles are still under debate (Hunt et al. 2007; McKenna & Farrell 2009; Misof et al. 2014; McKenna et al. 2015; Toussaint et al. 2017), we tentatively used the divergence times estimated by imposing fossils at superfamily level as our preferred time results.

We have added the considerations about using calibrations at the superfamily node in the

revised ms and also have mentioned the new dating analysis with a new supplementary figure (supplementary Fig. 9)

Minor things

Manuscript Title. Plural s for genes

Reply: gene -> genes

Line 118. Wrong ref number referred to

Reply: Wrong ref number has been corrected.

Line 169-171. This was showed already by ref 12

Reply: ref 12 has been added in citation.

Line 219-220. The first part of this sentence is showed, but that they underwent “rapid divergence after the extinction” is unclear.

Reply: This sentence was shorten to “These results indicated that Polyphaga originated in the Permian and survived through the End-Permian mass extinction.”

Line 241 “of” lacking

Reply: “division ecological niches” -> “division of ecological niches”

Line 266 a “t” lacking in plants

Reply: “t” was added.

Line 373 was should be is

Reply: “was” -> “is”.

Suppl table 4

Table heading: Sharing should be shared

Line 126 includes better than covers

Line 125 Hydrophiloidea misspelled

Reply: Modifications have been made as suggested and the misspelling has been corrected.

Suppl table 2

Is column PI Sites (Parsimony informative sites) important for anything in the study? Parsimony analysis is not used in the study, alignment length together with number of variable sites seems sufficient.

Reply: We have deleted the column PI Sites as suggested.

The study by Sharkey et al (Scient. Rep. 2017), should be mentioned, for instance when discussing the relationship between the four suborders, as they found the same as in the present study. It uses 358 gene clusters from translated transcriptome assemblies to infer a species phylogeny of

some 50 taxa of beetles including all four suborders, even though it has another focus than beetle phylogenetics.

Reply: Thank you for your reminder. This paper has been mentioned and cited in our revised ms.

Supplementary table 1 can be improved. First it is not useful for an overview to have it sorted by voucher number – better after taxonomic units. Second the locality column is inconsistent. For example country is not always given first but sometimes just QLD (for Queensland, Australia), “Thailand material” what is that?, Romania is misspelled, name of collector is often given without this being explained in header, same collector sometimes with sometimes without initials, etc

Reply: We have improved the supplementary table 1 as suggested: (1) the table is now sorted by taxonomic units; (2) the information of the locality column is clearer now; (3) we have added a column of collectors’ full name in the table.

Reviewer #2 (Remarks to the Author):

Review of “Evolutionary history of Coleoptera revealed by extensive sampling of gene and species”

This is a valuable paper on the phylogeny of one of the major lineages of life. Its value relative to other recent papers is the large number of nuclear protein-coding genes (NPC) used for a relatively large number of species. Its family level coverage (67% of living families) is not as extensive as McKenna et al. (94% of living families), and has some serious deficiencies, but its NPC sampling is much more extensive (95 NPCs versus 6 NPCs).

The paper is worth publishing, but it does have some critical problems, which should be corrected.

1. Identifications of specimens.

This is handled poorly, and yet it is a key aspect of any systematics paper.

Among the most concerning problems is the lack of identifications of many of the specimens. A full 69 of the species are not identified to genus, and frequently not to subfamily. I have never seen a systematics paper with anywhere near this level of “unknowns”. The 95 genes for each of those specimens will effectively be lost to the world – they won’t be able to be combined with any data in the future in any sensible way, except at the broadest levels. (Yes, eventually someone knowledgeable might examine the vouchers, but until that happens, the data have little value.) What does one do in the future with data from some unknown Dryopidae, or unknown Cryptophagidae? It can’t possibly be used for analyses within those families, and to the extent that some families are not monophyletic, it may not be usable at higher levels either. I realize that it is hard to identify beetles to genus in some parts of the world, but surely more can be done to identify them to a finer level than just “Cryptophagidae sp.”. If some of those belong to undescribed genera, then at least give an indication as to what they are close to, e.g., “New genus near X”. If the authors do not themselves have the coleopterological knowledge to identify specimens, they should reach out to experts in various groups to help with identifications.

This doesn’t just affect future use of the data, it affects a reader’s interpretations of the results. For example, consider the family Dryopidae. If someone were interested in its placement, knowing whether a diverse sample of dryopids are included in the analysis would be important. If the authors sampled dryopids representing the diversity of the family, then I would trust the results more than if only one small group of closely related dryopids were included. But all three dryopids included are listed simply as “gen. sp.”, so we have no idea whether they represent diverse dryopids or are extremely close.

Along a similar vein, it is not clear who identified many of the taxa. Without that knowledge, we cannot judge the quality of the identifications. E.g., Who identified the Myrabolia? Can we trust

the identification? The authors state that the collectors identified the specimens, but it is not always clear who the collectors were. In Supp Table 1, we can infer that a single word at the end of the Locality might be the last name of a collector, but this is not always clear. And if it is clear, it can require special knowledge to know of whom they speak. For example, there are numerous specimens with the word “Monteith” at the end of the locality data. I presume this is Geoff Monteith, but how are other readers to know this? The identifier should be listed, clearly, for all specimens. And not just their last names, but their full names, so that readers have the information they need to judge the quality of the identifications.

Reply: Thank you for your comments. There are 69 specimens not identified to genus. We have done our best to carefully recheck the vouchers of these 69 specimens. 52 of 69 specimens now have been identified to genus. About your concerns, the two samples of the family Cryptophagidae have been identified to CSR015: *Micrambina sp.* and INB192: *Curelius sp.* The three samples of the family Dryopidae have now been identified to CSR019: *Helichus sp.*, INB166: *Pachyparnus sp.*, and INB194: *Helichus sp.* Now there are still 17 specimens that cannot be identified to genus because these voucher specimens were partly damaged during the DNA extraction process and lacked of enough details for more accurate identification. We hope the reviewer can understand that we have tried our best with identifications.

In addition, our coauthors Li Yun (China) and Adam Ślipiński (Australia) identified most of beetle specimens used in this study. We have added a collector/identifier column in the supplementary table 1 to show the collector/identifiers' full names. We have rephrased the sentence within the Materials and methods to “All specimens derived from the Biological Museum of Sun Yet-Sen University, China and Australian National Insect Collection, CSIRO, in Canberra, Australia were marked with unique numbers. The detailed information of taxonomy, locality, collector/identifier of specimens are provided in Supplementary Table 1.” in the revised ms. We hope that, with a more complete information, our beetle sequence data can be more useful in future beetle phylogenetic studies.

2. Some major results are based upon very few species, and could easily be artifacts resulting from long branch attraction.

A major result of the paper, perhaps the major result, are the inferred relationships of the four suborders. However, included in their analysis is one representative of Archostemata and one of Myxophaga. This is in notable contrast to McKenna et al. 2015, which had in their analysis seven species in three families of Archostemata, and six species in all four families of Myxophaga. Having only a single species of Archostemata and only one of Myxophaga makes these taxa prone to long-branch attraction, and thus their placement is much less certain. This is especially problematic given that the relationships of the four suborders are in part an issue of rooting: the nine proposed hypotheses of subordinal relationships listed in Table 1 represent only three unrooted hypotheses. Because of the long phylogenetic distance to the outgroups, the position of the root of Coleoptera is especially sensitive to long-branch attraction; having only one archostematan and one myxophagan makes it even more so.

Another example are the claims about the paraphyly of Hydradephaga. This paraphyly results from the position of Haliplidae and Gyrinidae. However, these divergent families are represented by only a single species each, and thus they are on long branches. These long branches are more likely to be attracted to each other, and are more likely to be artificially attracted to the base of Adephaga.

I realize it would be difficult to add more Myxophaga and Archostemata at this point. If this cannot be done, then the authors need to be explicit about the problems of having only single species, including potential long branch attraction, and should therefore be less bold in their claims.

Reply: For the relationships of the four suborders, to explore whether our result were influenced by LBA, we made a small version of our original data set and used the CAT+GTR+ Γ site-heterogeneous mixture model (Lartillot et al. 2013) that are generally less sensitive to LBA to reconstruct the phylogeny for the four suborders. We still got the same topology with strong support (see the figure below). This result show that the close relationships between Archostemata and Myxophaga is unlikely caused by LBA.

However, indeed, our AU test analyses cannot reject some alternative topologies. So the answer is really not so solid. So we added the following sentences in the end of the corresponding paragraph to make our claims less strong. "It should be noted that our study included only a single species of Archostemata and only one of Myxophaga, which makes these taxa prone to long-branch attraction (LBA). Therefore, the relationships among Adephaga, Myxophaga, and Archostemata reported here should still be considered as tentative and needed validation by future studies."

For the claims about the paraphyly of Hydradephaga, we also used the CAT+GTR+ Γ model to reconstruct the tree and found that the position of Haliplidae and Gyrinidae inferred by the CAT+GTR+ Γ model were different and weakly supported (see the figure below). So we added the following sentences in the end of the corresponding paragraph to make our claims less strong. "Therefore, the phylogeny of Adephaga still remained ambiguous. Given that 95 genes were used in this study, resolving the internal relationships of Adephaga may require using much bigger data sets (i.e., transcriptomes)."

23,802 amino acids, 60 taxa, CAT+GTR+GAMMA model. Branch supports are posterior probabilities

3. Timescale

The authors state “We attribute the greater precision [of dating estimates] to the size of our dataset (71 kb), which exceeds those of previous studies (~8.4 kb at most) by at least eight fold”. It’s not clear to me why this would necessarily follow, as there are two differences between their analysis and McKenna et al.: the amount of DNA data and different age constraints. The authors claim, if true, would be very interesting. For these reasons it would be very valuable if the authors not just speculate, but actually test this. If they redo the analysis using their 95 genes but the exact same age constraints as used by McKenna et al., and they still had the increased precision, then their claim would be corroborated; if they had the same precision as McKenna et al, then the result could be a result of the different age constraints.

Reply: Thanks for your suggestion. We have redone the time analyses using our 95 genes but the exact same age constraints as used by McKenna et al. and Toussaint et al. We still observed the increased precision of the dating results. We have mentioned this new result in the revised ms with a new supplementary figure (Supplementary Fig. 10).

Methods:

1. The authors should provide some argument as to why they did not include representatives of the sister group of beetles, Strepsiptera, as an outgroup to beetles.

Reply: We did not include representatives of the sister group of beetles, Strepsiptera, as an outgroup to beetles, because among the 95 genes used in this study, only 44 could find orthologous sequences in the published Strepsiptera genome (missing data > 50%). We have added this argument into the revised ms.

2. Why was the data set divided into 10 partitions based upon rate? Why rate? Why 10? A more coherent method is to use a program such as PartitionFinder to determine the optimal partitioning of data. This will take into account factors that are more critical to model fitting than is simple rate.

Reply: Because our data set includes about 400 species and more than 20,000 positions of amino acids, using PartitionFinder to determine the optimal partitioning scheme and corresponding best-fit models is computational challenging. On the other hand, many genes in our data set are short and do not have enough informative sites to accurately calculate model parameters. Binning genes into "supergenes" is a statistical technique that can account for sampling error by increasing signal-to-noise ratio, and it has been applied in phylogenetics recently (Bayzid and Warnow 2013; Betancur-R et al. 2014; Mirarab et al. 2014). So we used data binning strategy to partition our data and inferred the best-fit model of each bin partition using ProtTest. We divided genes into bins based on rates because genes with similar rates are expected to have similar evolutionary models, which is prone to decrease the risk of model misspecification.

We divided the protein data set into 10 binned partitions due to computational effectiveness. We also did additional phylogenetic analyses using different partitioning schemes. The phylogenies inferred from unpartitioned and 95-gene-partitioned dataset are almost identical to that inferred from 10-bin-partitioned dataset, except several nodes with negligible supports (Supplementary Fig. 12). This result showed that different partitioning schemes have little influence to the final phylogenetic results. We have mentioned the above results in the revised ms.

Some minor issues:

The paper Baca et al. 2017 ("Ultraconserved elements show utility in phylogenetic inference of Adephaga (Coleoptera) and suggest paraphyly of 'Hydradephaga' ") should be cited, and its results incorporated into the discussion.

Reply: Thank you for your reminder. This paper has been mentioned and cited in our revised ms.

The authors state "We estimated that the last common ancestor of extant beetles occurred during the earliest Permian at 297 mya (95% CI 291 – 304), which is earlier than the Early-Late Permian origin (285 – 253 mya) estimated by Hunt et al. and McKenna et al. but later than that of Toussaint et al. (~ 333 mya) (Fig. 3b). The earliest Permian origin of Coleoptera suggested in this study is consistent with the beetle fossil record, because the oldest definite beetle fossil (Coleopsis archaica) dates back to the earliest Permian at approximately 295 million years ago" . The logic of the last sentence is not clear to me. Coleopsis is part of the stem lineage of

Coleoptera, and for this reason could occur before or after the origin of the common ancestor of extant beetles; in what way is the age of Coleopsis relevant for the age of the common ancestor of extant beetles?

Reply: This confusing sentence "The earliest Permian origin of Coleoptera suggested in this study is consistent with the beetle fossil record, because the oldest definite beetle fossil (Coleopsis archaica) dates back to the earliest Permian at approximately 295 million years ago. " has been deleted in the revised ms.

Spelling of authors name in reference 16 is incorrect.

Reply: The misspelling has been corrected in the revised ms.

Reference 30 has an incorrect title

Reply: The title of the reference has been corrected in the revised ms.

Reviewer #3 (Remarks to the Author):

The authors report what is likely to be the most complete molecular dataset so far published for Coleoptera, and reconstruct a well resolved, well supported calibrated phylogeny of the Order. The methodological details were published in an associated paper (see below), but the results seem robust and the conclusions well justified. In my opinion the paper is acceptable for publication, with only some minor modifications.

A relatively high proportion of taxa is not identified not even at the genus level. The authors provide a substantial genetic resource for many taxa, but its usefulness is greatly diminished if the identification is only at the family level. In some cases it may be complex to arrive at a generic identification, but for others this seems relatively simple, as there are many genera-poor families for which the identification should be possible.

Reply: Thank you for your comments. There are 69 specimens not identified to genus. We have done our best to carefully recheck the vouchers of these 69 specimens. 52 of 69 specimens now have been identified to genus. Now there are still 17 specimens cannot be identified to genus because these voucher specimens were partly damaged during the DNA extraction process and lacked of enough details for more accurate identification. The new information is provided in the revised supplementary table 1. We hope the reviewer can understand that we have tried our best with identifications.

In the Abstract, the sentence "sister to all other polyphagans, excluding Scirtoidea, Derodontidae and Elateriformia" (as well as similar sentences in the text) may be confusing, as it suggests a more basal position for Nosodendrinae. Would it be better something like "sister to Staphyliniformia, Bostrichiformia and Cucujiformia"?

Reply: Thank you for your suggestion. In the revised ms, we have rephrased the sentence in the Abstract and in the text as suggested.

The sentence " nine hypotheses regarding the relationships among four suborders of beetles (Fig. 1) have been proposed in recent decades, although none have received strong support^{11,16,18 – 23.}" is misleading, as in some cases topologies were strongly supported, only with a poor taxonomic sampling (e.g. Pons et al.).

Reply: Thank you for your reminder. The sentence has been rephrased to "nine hypotheses regarding the relationships among four suborders of beetles (Fig. 1) have been proposed in recent decades, but most of them did not receive strong support" in the revised ms.

"The inconsistency between phylogenetic relationships and variable age estimates is probably because of the small number of molecular markers traditionally used for beetle phylogenetics." - the effect of a poor sampling must also be considered. In previous phylogenies, when the amount of sequence was enough, the sampling was very poor (e.g. Misof et al.), or when the sampling was good, the amount of sequence was minimal (e.g. Hunt et al.). However, the large differences with the dating of Toussaint et al. may require some additional explanation. Could they be because of the different use of fossils? Some methodological differences in the calibration? "Small number of markers" does not seem to be reason enough here.

Reply: Thank you for your comments. Now the heading sentence is modified to "It is well known that both taxon sampling and gene sampling can affect the accuracy of phylogenetic reconstruction. In previous beetle phylogenetic studies, when the amount of sequence was large (e.g., using ribosomal protein genes extracted from EST data (Hughes et al. 2006), whole mitochondrial genomes (Crampton-Platt et al. 2015; Timmermans et al. 2016), or transcriptome data (Misof et al. 2014; Peters et al. 2014)), the taxon sampling was small, or when the taxon sampling was large, the amount of sequence was small (e.g., three genes: ~3000 nt (Hunt et al. 2007), four genes: 6600 nt (Bocak et al. 2014), and eight genes: 8377 nt (McKenna et al. 2015))"

Now we do not discuss the time in this paragraph, so it is no need to explain the differences of times. To supplement this, in the discussion section, we have pointed out that the divergence times of beetle evolution are sensitive to the fossil calibration points.

The choice of the partition scheme needs more explanation. Why ten partitions according to the p distances of the genes? What effect could do the use of say 9, or 11 partitions? Seems an arbitrary choice, which either needs more justification, or to be shown that the actual number is irrelevant within a reasonable range.

Reply: Many genes in our data set are short and do not have enough informative sites to accurately calculate model parameters. Binning genes into "supergenes" is a statistical technique that can account for sampling error by increasing signal-to-noise ratio, and it has been applied in phylogenetics recently (Bayzid and Warnow 2013; Betancur-R et al. 2014; Mirarab et al. 2014). So we used data binning strategy to partition our data and inferred the best-fit model of each bin partition using ProtTest. We divided genes into bins based on rates because genes with similar rates are expected to have similar evolutionary models, which is prone to decrease the risk of model misspecification.

We divided the protein data set into 10 binned partitions due to computational effectiveness. We also did additional phylogenetic analyses using different partitioning schemes. The phylogenies inferred from unpartitioned and 95-gene-partitioned dataset are almost identical to that inferred from 10-bin-partitioned dataset, except several nodes with negligible supports (Supplementary Fig. 12). This result showed that different partitioning schemes have little influence to the final phylogenetic results. Therefore, we don't think using 9 or 11 partitions will change our phylogenetic results. We have mentioned the above results in the revised ms.

"For the nucleotide data set, we used a Perl script (Degen_v1_4.pl; <http://www.phylotools.com>) that degenerates the first and third codons encoding the same amino acid to minimize heterogeneous base composition⁵⁹" - the authors should provide more details. RAxML, Beast and other software cannot deal with ambiguous positions - they should be either missing (or N) or defined. Were all 1st and 3rd codon positions just transformed to "N", i.e. effectively deleted from the analyses?

Reply: The Perl script Degen_v1_4.pl is designed to fully degenerate codons of sequences with the synonymous change, for example, 2-fold degenerate sites in third codon are degenerated to "R" or "Y", 4-fold degenerate codons with the third positions are transformed to "N", Leu codons are degenerated to "YTN". RAxML will treat "N" as missing and the column of the third codon position with undetermined "N" is removed automatically.

We have modified the sentence in the revised ms to "For the nucleotide data set, we used a Perl script (Degen_v1_4.pl; <http://www.phylotools.com>) to degenerate nucleotides to IUPAC ambiguity codes for the first and third codon positions, an extension of RY coding, which can reduce the effect of nucleotide compositional heterogeneity."

There is no information of the primers used - or even the full name of the genes - other than a reference to an associated work (Che et al.). It would be useful to have more data, at least in the supplement.

Reply: We have added the primer information of 95 genes into the supplementary table 2 as suggested.

In Table 1 it would be useful to include the numbers of the topologies in Fig. 1, to readily see which is which

Reply: Table 1 has been added the numbers of the topologies in Fig. 1 as suggested.

Should be "Drosophilidae" not "Drosophylidae" (Supp. Table 3) and "Anacaena" not "Anaceana" (Fig. 2, supplements)

Reply: We have corrected these spelling errors in all figure and supplements.

The alignments are said to be deposited in Dryad: I assume the sequences are also to be deposited in genbank?

Reply: We had tried to submit our sequences to GenBank. However, the genbank staffs found that there are intronic sequences in some of our sequences and told us they would not accept these kind of data unless we can precisely annotate these sequences. We have tried to give

annotations to these sequences. But after many tries, we still could not meet the requirements of Genbank. So we have to deposit our sequences in figshare.

REVIEWERS' COMMENTS:

Reviewer #3 (Remarks to the Author):

The authors have addressed most of the suggestions of the Referees, and the ms is now in a better shape. There are, however, some issues that still need to be addressed:

Table S1: The legend still says that specimens were sorted by voucher number.

371-376: The RY coding was used for the nucleotide dataset, but it is still no clear to me the effect of this transformation in the different phylogenetic methods used. RAXML can handle RY coding (although for the answer of my previous comment I assume that ambiguous positions were treated as missing), but some of the ML and Bayesian software used do not handle RY coding, and what they effectively do is to treat ambiguous positions as missing data. If so this is a rather dramatic effect that should be clarified adequately in the ms, noting how the different software used deal with the RY coding and the potential effect. This is not crucial as the preferred topology is that obtained with AA anyway, but should be clarified.

351-353: It is somewhat disturbing that the sequence data could not be submitted to Genbank. In the text it is said that introns were removed and the sequences checked so that they were properly translated (as some analyses used the AA sequence). If so there should be no problem in submitting the sequences, but in the response letter it is said that sequences could not be submitted due to the presence of introns (which should not be there?). Genbank staff are usually highly competent, so I'm inclined to think that there were still some issues with the data - which, as said, is disturbing. In many journals submission of the raw sequences to Genbank is compulsory for this type of studies, do not know how NC will handle this situation.

line 136: It is said that "internal relationships of Adephaga may require using much bigger data sets (i.e., transcriptomes)". The fact is that the sampling of Adephaga is rather poor (as well as that of Myxophaga and Archostemata), with a few families absent (Hygrobiidae, Amphizoidae, Meruidae, Aspidytidae) - event if all these are species poor families, it may be more determinant to expand the sampling than to enlarge the dataset.

Finally, although as far as I can ascertain the English is generally correct, the ms may benefit from a careful reading to smooth some sentences and in general to improve the expression. Two examples are included below:

line 218-219: "for the twelve nodes in the beetle tree" - certainly there are more than 12 nodes in the tree, changes to "12 selected nodes", or delete "the"

line 225: "the increased precision should mainly attributes" - should be mainly attributed?

Response to reviews (in blue)

REVIEWERS' COMMENTS:

Reviewer #3 (Remarks to the Author):

The authors have addressed most of the suggestions of the Referees, and the ms is now in a better shape. There are, however, some issues that still need to be addressed:

Table S1: The legend still says that specimens were sorted by voucher number.

Reply: The legend of the Supplementary Data 1 (originally named Table S1) has been rephrased to "Information of all samples used in this study, including taxonomy, voucher number and collection information. "/"=not applicable, "-"= no information."

371-376: The RY coding was used for the nucleotide dataset, but it is still no clear to me the effect of this transformation in the different phylogenetic methods used. RAXML can handle RY coding (although for the answer of my previous comment I assume that ambiguous positions were treated as missing), but some of the ML and Bayesian software used do not handle RY coding, and what they effectively do is to treat ambiguous positions as missing data. If so this is a rather dramatic effect that should be clarified adequately in the ms, noting how the different software used deal with the RY coding and the potential effect. This is not crucial as the preferred topology is that obtained with AA anyway, but should be clarified.

Reply: The Perl script Degen_v1_4.pl. transforms the codons specify an amino acid into degenerated codons (having ambiguous characters). For instance, codons specify glutamic acid is transformed into "GAR"; codons encoding the amino acid arginine is changed to "MGN". As a result, all three-base pair codons were compressed to degenerated codons (ambiguous nucleotides) according to IUPAC notation. Moreover, the amino acid serine is not transformed into "WSN", but specified by "TCN" (Ser1) or "AGY" (Ser2). Therefore, the degenerated nucleotide dataset contained gap ("-") or missing ("N"), and ambiguous characters ("R","Y","H","M").

Ambiguous characters are supported in the RAXML and IQ-TREE (their manuals clearly state this point) and will not treated as missing data. Only columns that contains only gaps or "N" will be treated as missing data. We redid our DNA ML analyses with both RAXML and IQTREE after replacing all ambiguous characters (RYHM) into missing characters (N). The resulting topologies are nearly identical, but support values generally decrease. This means that ambiguous characters ("R","Y","H","M") can be supported by RAXML and IQTREE and are informative in ML analyses.

We have added a sentence to clarify this in the ms. "The degenerated nucleotide dataset makes synonymous changes largely invisible but non-synonymous changes largely intact, which can be supported in RAXML and IQ-TREE analyses."

351-353: It is somewhat disturbing that the sequence data could not be submitted to Genbank. In the text it is said that introns were removed and the sequences checked so that they were

properly translated (as some analyses used the AA sequence). If so there should be no problem in submitting the sequences, but in the response letter it is said that sequences could not be submitted due to the presence of introns (which should not be there?). Genbank staff are usually highly competent, so I'm inclined to think that there were still some issues with the data - which, as said, is disturbing. In many journals submission of the raw sequences to Genbank is compulsory for this type of studies, do not know how NC will handle this situation.

Reply: We have submitted the assembled sequences to GenBank. Now every gene sequence has its GenBank accession number. All GenBank accession numbers are now integrated in a excel table (Supplementary Data 3). Researchers can download these sequences according to their GenBank accession numbers.

line 136: It is said that "internal relationships of Adephaga may require using much bigger data sets (i.e., transcriptomes)". The fact is that the sampling of Adephaga is rather poor (as well as that of Myxophaga and Archostemata), with a few families absent (Hygrobiiidae, Amphizoidae, Meruidae, Aspidytidae) - event if all these are species poor families, it may be more determinant to expand the sampling than to enlarge the dataset.

Reply: We have rephrased this sentence to "Given that our sampling of Hydradephaga was insufficient, recovering robust internal relationships of Adephaga may require sampling more aquatic adephagans from families Amphizoidae, Aspidytidae, Hygrobiiidae, and Meruidae."

Finally, although as far as I can ascertain the English is generally correct, the ms may benefit from a careful reading to smooth some sentences and in general to improve the expression. Two examples are included below:

line 218-219: "for the twelve nodes in the beetle tree" - certainly there are more than 12 nodes in the tree, changes to "12 selected nodes", or delete "the"

line 225: "the increased precision should mainly attributes" - should be mainly attributed?

Reply: Thank you for your suggestions. We have modified these sentences as you suggested in the ms. The revised manuscript has been carefully viewed by a native English speaker and we have tried our best to improve the English.